# CuS-^131^I-PEG Nanotheranostics-Induced “Multiple Mild-Hyperthermia” Strategy to Overcome Radio-Resistance in Lung Cancer Brachytherapy

**DOI:** 10.3390/pharmaceutics14122669

**Published:** 2022-11-30

**Authors:** Yanna Cui, Hui Yan, Haoze Wang, Yongming Zhang, Meng Li, Kai Cui, Zeyu Xiao, Liu Liu, Wenhui Xie

**Affiliations:** 1Department of Nuclear Medicine, Shanghai Chest Hospital & Department of Pharmacology and Chemical Biology, Translational Medicine Collaborative Innovation Center, Shanghai Jiao Tong University School of Medicine, Shanghai 200025, China; 2College of Chemistry and Materials Science, Shanghai Normal University, Shanghai 200233, China

**Keywords:** PEGylated nanotheranostics, brachytherapy, multiple mild-hyperthermia, tumor hypoxia, radio-resistance

## Abstract

Brachytherapy is one mainstay treatment for lung cancer. However, a great challenge in brachytherapy is radio-resistance, which is caused by severe hypoxia in solid tumors. In this research, we have developed a PEGylated ^131^I-labeled CuS nanotheranostics (CuS-^131^I-PEG)-induced “multiple mild-hyperthermia” strategy to reverse hypoxia-associated radio-resistance. Specifically, after being injected with CuS-^131^I-PEG nanotheranostics, tumors were irradiated by NIR laser to mildly increase tumor temperature (39~40 °C). This mild hyperthermia can improve oxygen levels and reduce expression of hypoxia-induced factor-1α (HIF-1α) inside tumors, which brings about alleviation of tumor hypoxia and reversion of hypoxia-induced radio-resistance. During the entire treatment, tumors are treated by photothermal brachytherapy three times, and meanwhile mild hyperthermia stimulation is conducted before each treatment of photothermal brachytherapy, which is defined as a “multiple mild-hyperthermia” strategy. Based on this strategy, tumors have been completely inhibited. Overall, our research presents a simple and effective “multiple mild-hyperthermia” strategy for reversing radio-resistance of lung cancer, achieving the combined photothermal brachytherapy.

## 1. Introduction

Nowadays, lung cancer is the leading cause of cancer mortality in both men and women worldwide. [1]. Radiotherapy is a mainstay of treatment for lung cancer that is delivered by external or internal rays [2,3]. External radiotherapy based on high-energy radiation of X-rays or γ-rays can effectively destroy cancer cells [4], but in the meantime it causes severe damage to normal thoracic organs that are penetrated by the radioactive rays [5]. In contrast, internal radiotherapy, especially brachytherapy, has been widely applied to lung cancer treatment owing to its prominent advantages of precisely localized radiation at a short range and lower toxicity to normal thoracic organs [6]. In addition, brachytherapy is also a salvage treatment for difficult patients for whom traditional treatments (e.g. chemotherapy, immunotherapy, surgery and external radiotherapy) have failed [7]. During brachytherapy, the cancer cell-damaging effect is much stronger under well-oxygenated conditions [8,9]. However, due to the hypoxic state in solid tumors, brachytherapy efficiency is severely limited by hypoxia-induced radio-resistance.

Mild hyperthermia in local tumor tissues is one alternative strategy for alleviating tumor hypoxia. Mild hyperthermia induced by photothermal agents can slightly increase the tumor temperature, to simply boost the high blood flow into the tumor, effectively increase the local oxygen level, and controllably reshape the hypoxic tumor microenvironment [10]. However, tumor tissues normally recover their hypoxic state at about 24 h after the stimulation of mild hyperthermia, again leading to the hypoxia-associated radio-resistance [11]. With the deepening of research in the nano-material field, multiple mild-hyperthermia stimulation based on photothermal nano-agents will realize the repeated alleviation of tumor hypoxia. For example, CuS photothermal nano-agents have a stronger photothermal stability that is ideally suited for repeated NIR-irradiation and better photothermal conversion capability that is useful for achieving tumor hyperthermia [12,13,14,15]. Moreover, radiopharmaceutical ^131^I is easily embedded into the crystal defect sites of CuS by virtue of the strong binding energy between Cu^2+^ and I^−^ ions [16,17]. After being labeled with ^131^I, CuS-^131^I nanoparticles would perform the excellent photothermal brachytherapy, once the tumor hypoxic environment is reversed by mild hyperthermia under NIR irradiation. In addition, PEG as one extensively used hydrophilic polymer can be easily coated onto the surface of nano-drugs or used as nano-carriers due to its rich functional group. PEGylation on the surface of nanoparticles could enhance their tumor accumulation through enhanced and permeation-retention (EPR) effect [18]. That is, higher tumor accumulation of radiopharmaceuticals (e.g. ^131^I) can be guaranteed by PEG coating. However, there have been no reports about PEGylated CuS-^131^I nanoparticles to be used to repeatedly regulate tumor hypoxia through multiple mild-hyperthermia stimulation.

Herein, we have prepared a kind of biodegradable PEGylated nanotheranostics (CuS-^131^I-PEG) for long-term alleviating of tumor hypoxia and reversing radio-resistance through a “multiple mild-hyperthermia” strategy. In this nanotheranostics, radionuclide ^131^I emitting stronger beta and gamma rays served as a radio-pharmaceutical. ^131^I can be easily embedded into crystal defect sites of photothermal agents (CuS) to obtain ^131^I-labeled CuS (CuS-^131^I) [19]. Thiol-poly (ethylene glycol) as a biodegradable polymeric matrix is modified on the surface of CuS-^131^I nanoparticles through high binding energy between Cu and -SH [20]. PEGylation of CuS-^131^I can enhance the stability of ^131^I and prolong its retention time in tumor tissues according to radioactive meter and SPECT/CR imaging. After the intratumoral injection of CuS-^131^I-PEG nanotheranostics, tumors on mice are exposed to the first NIR laser to induce mild hyperthermia. This mild hyperthermia can enhance oxygen levels in tumors and decrease HIF-1α expression so as to alleviate tumor hypoxia and reverse hypoxia-induced radio-resistance. Meanwhile, mild hyperthermia induced by the first NIR laser irradiation also has promoted the uniform intra-tumoral distribution of CuS-^131^I-PEG nanotheranostics. Based on tumor hypoxic alleviation and intratumoral uniform distribution of nanotheranostics, the combined photothermal brachytherapy under the second NIR laser irradiation has been achieved. In this study, the combined photothermal brachytherapy is conducted three times. Mild-hyperthermia stimulation is carried out two hours in advance of each combined therapy. These three mild-hyperthermia stimulations are defined as the “multiple mild-hyperthermia” strategy. Tumors in the combined group with the “multiple mild-hyperthermia” strategy have been successfully inhibited compared with control group without mild-hyperthermia. Overall, our research firstly develops the “multiple mild-hyperthermia” strategy for repeatedly alleviating tumor hypoxia, reversing hypoxia-associated radio-resistance and finally achieving the combined photothermal brachytherapy of lung cancer, which has great clinical application prospects.

## 2. Materials and Methods

### 2.1. Materials

Copper chloride (CuCl_2_), sodium citrate, sodium iodine (NaI) and sodium sulfide (Na_2_S) were purchased from Sino-pharm Chemical Reagent Co., Ltd. (Shanghai, China). Thiol PEG (MW. 5000), PEG-SH was purchased from Yeasen Biotech Co., Ltd (Shanghai, China). Iodine-131 (Na^131^I) was supplied by Shanghai atom kexing pharmaceuticals Co., Ltd. (Shanghai, China). RPMI 1640 cell culture medium, Trypsin-EDTA Solution, penicillin-streptomycin (P/S) solution and fetal bovine serum (FBS) were obtained from Gibco (Grand Island, NY, USA). Cell Counting Kit-8 (CCK-8), HIF-1α rabbit polyclonal antibody and Alexa Fluor 647-labeled Goat Anti-Rabbit IgG (H + L) were purchased from Beyotime Biotechnology Co., Ltd. (Shanghai, China). HypoxyprobeTM-1 Plus Kit was purchased from Hypoxyprobe, Inc. (Burlington, MA, USA).

### 2.2. Preparation of PEGylated Nanotheranostics (CuS-^131^I-PEG)

Preparation process of CuS-^131^I-PEG nanotheranostics contained three steps. Firstly, ultra-small CuS nanoparticles were synthesized by our previous method [21]. In brief, CuCl_2_ (13.45 mg) and citrate sodium (20 mg) were added in ddH_2_O (100 mL) under stirring. Then, Na_2_S (8.68 mg) was dissolved into the above solution. The reaction solution was heated to 90 °C and stirred for 15 min. The resultant citrate-modified CuS nanoparticles were collected and stored at 4 °C. Secondly, to label radionuclides, the above obtained CuS nanoparticles were mixed with Na^131^I under magnetic stirring for 2 h at room temperature. ^131^I-labeled CuS (CuS-^131^I) was purified by centrifuge filtration (MWCO: 10 kDa) to remove free Na^131^I. Radioactivity and radio-labeling efficiency of CuS-^131^I were measured by a radioactivity meter (CRC-25R, CAPINTEC Inc., Pittsburgh, PA, USA). Finally, PEGylated CuS-^131^I (CuS-^131^I-PEG) was prepared by mixing PEG-SH and CuS-^131^I. CuS-^131^I-PEG was also purified by centrifuge filtration (MWCO: 10k Da) to remove free PEG-SH. The obtained CuS-^131^I-PEG was stored at 4 °C for the following studies.

In order to optimize the iodine amount and PEG coating amount, radionuclide Na^131^I was substituted with normal NaI in the above processes. As shown in Appendix A, the mass ratio of CuS to NaI was changed from 5:1 to 20:1. Meanwhile, the PEG coating amount was increased from 0.5 mg to 10 mg when CuS was kept at 3.36 mg.

### 2.3. Characterization of PEGylated Nanotheranostics (CuS-^131^I-PEG)

Morphology of CuS-^131^I-PEG nanotheranostics was detected through transmission electron microscopy (FEI-TEM, US, 200 kV of accelerating voltage). Hydrodynamic size and zeta potential were measured by dynamic light scattering technique (Zetasizer Nano ZSP, Malvern, UK). Absorption spectra were observed by UV-Vis-NIR spectrophotometer (EV300, Hitachi, Santa Clara, CA, USA). Radiolabeling efficiency, storage stability and serum stability of CuS-^131^I-PEG were evaluated by radioactivity meter. Photothermal conversion properties were investigated by 808 nm NIR laser at a power intensity of 2.8 W/cm^2^, and temperature changes were monitored by infrared thermal camera (FLIR Ax5). Intratumoral diffusion and distribution of ^131^I-CuS-PEG was observed by multimodal small animal photoacoustic imaging system (VEVO LAZR-X, FujiFilm VisualSonics Inc., Bothell, WA, USA). Radioactivity of ^131^I-CuS-PEG and free ^131^I in vivo was collected by nanoScan^®^ SPECT/CT imaging system, and the obtained data were reconstructed by Nucline Nanoscan program, Ver.3.04 (Mediso Ltd., Budapest, Hungary).

### 2.4. Cell Culture and Animal Model

A549 cells were cultured in RPMI 1640 medium containing 10% FBS and 1% P/S in the standard incubator (5% CO_2_) at 37 °C for the following in vitro and in vivo experiments.

Balb/c mice, six weeks old, were purchased from Shanghai SLAC Laboratory Animal Co., Ltd. (Shanghai, China). A549 cells (2 × 10^6^ cells/mouse) were subcutaneously injected into the dorsum area of mice. Tumors sized 125~150 mm^3^ were used for animal imaging and therapeutic studies. All animal experiments were conducted according to the protocol approved by the Animal Care Committee of the Shanghai Jiao Tong University School of Medicine.

### 2.5. In Vitro Serum Stability of PEGylated Nanotheranostics (CuS-^131^I-PEG)

CuS-^131^I-PEG nanotheranostics were dispersed in 1 mL of 10% fresh mouse serum and incubated at 37 °C or room temperature for 16 h, respectively. After incubation for 16 h, CuS-^131^I-PEG was ultra-filtrated by centrifuge at 5000 rpm for 15 min. The dissociated ^131^I in the lower filtrate was detected by radioactivity meter to evaluate the serum stability of CuS-^131^I-PEG. In the control group, CuS-^131^I-PEG nanotheranostics were incubated at PBS.

### 2.6. In Vitro Cytotoxicity of PEGylated Nanotheranostics (CuS-^131^I-PEG)

A549 cells were firstly seeded in 96-well micro-plates at a density of 5000 cells per well in a 100 μL cell culture medium. After 24 h, the A549 cells were randomly divided into two groups and co-incubated with CuS-^131^I-PEG nanotheranostics for 24 h and 48 h, respectively. Two groups (n = 3 wells/group) were as follows: (i) Combined group of photothermal brachytherapy under normal cell culture conditions. After being co-incubated with CuS-^131^I-PEG nanotheranostics, the A549 cells were irradiated by 808 nm NIR laser for 5 min. The temperature of cell culture medium was raised to about 42 °C, to achieve the combined photothermal brachytherapy. (ii) Combined therapy group at hypoxic condition. Cobalt dichloride (100 μM) as a hypoxic inducer was co-incubated with the A549 cells in advance, to achieve hypoxic A549 cells. Twelve hours later, CuS-^131^I-PEG nanotheranostics with different concentrations were added to the hypoxic A549 cells and irradiated by 808 nm NIR laser for 5 min. Incubating free CuS-I-PEG nanoparticles with A549 cells were set as a control group. Finally, cell viabilities were evaluated by cell counting kit-8 assay (CCK-8).

### 2.7. In Vivo Photothermal Effect of PEGylated Nanotheranostics (CuS-^131^I-PEG)

To study the photothermal effect of CuS-^131^I-PEG, 10 μL of CuS-^131^I-PEG nanotheranostics were injected into A549 tumors on mice. Then, tumor regions were irradiated by 808 nm NIR laser with a laser power of 2.8 W/cm^2^ for 15 min, and the light-induced thermal effect was monitored by infrared thermal camera in the specific time interval. Meanwhile, we monitored the irradiation process and maintained the temperature in the range of 25~45 °C to avoid overheating. In control group, A549 tumors were injected with 10 μL of PBS.

During the photothermal effect study, the mice were anesthetized with isoflurane inhalation (2%) and kept under anesthesia by isoflurane inhalation (1~1.5%). After being anesthetized, the mice were placed on a heating pat to keep the body warm and have them breathe normally.

### 2.8. In Vivo Photoacoustic Imaging

A photoacoustic imaging technique was used to investigate the diffusion and distribution of CuS-I-PEG nanotheranostics after mild hyperthermia stimulation. A549 tumor-bearing mice were firstly anesthetized with isoflurance inhalation (2%) and then intratumorally injected with CuS-I-PEG nanotheranostics (10 μL). Then, tumors were irradiated by 808 nm NIR laser with a power of 1.5 W/cm^2^ so that temperature in tumors was raised to 39~40 °C and kept for 15 min. The intratumoral distribution of CuS-I-PEG nanotheranostics was monitored by photoacoustic imaging system at 30 min, 1 h, 2 h post-injection. Non-irradiated mice were considered as control group.

### 2.9. In Vivo SPECT/CT Imaging

Distribution and metabolic process of ^131^I in vivo were evaluated by a SPECP/CT imaging system. Firstly, mice were anesthetized by isoflurance inhalation (2%). Secondly, mice with 125~150 mm^3^ of A549 tumor were randomly separated into two groups (n = 3 mice/group): (i) intratumoral injection of free ^131^I solution; (ii) intratumoral injection of CuS-^131^I-PEG nanotheranostics with mild hyperthermia under 808 nm NIR laser. After being injected with different solutions, mice were scanned by SPECT/CT imaging system at 0.5 h, 1.5 h, 3 h, 4.5 h, 6 h post-injection, respectively. Finally, the collected data were reconstructed by Nucline Nanoscan program, Ver.3.04 (Mediso Ltd., Hungary).

### 2.10. Immunofluorescence Imaging

Hypoxic states (e.g., hypoxic level and HIF-1α expression) in solid tumors were evaluated by immunofluorescent staining technique. In brief, A549 tumors on mice were randomly separated into four groups (n = 3 mice/group): (i) Two-time NIR laser irradiation group. After being injected with CuS-^131^I-PEG nanotheranostics, tumors were irradiated by 808 nm NIR laser with a power of 1.5 W/cm^2^ at 15 min and 22 h post-injection, respectively. (ii) Single NIR laser irradiation group. After being injected with CuS-^131^I-PEG nanotheranostics, tumors were only irradiated by 808 nm NIR laser with a power of 1.5 W/cm^2^ for 15 min only at 15 min post-injection. (iii) Non-irradiated group. Tumors were not irradiated by NIR laser after intratumoral injection of CuS-^131^I-PEG nanotheranostics. (iv) blank control group. Tumors were left untreated. Finally, tumors were surgically excised at 2 h and 24 h post-injection to make frozen sections, respectively.

To detect hypoxic level, tumor hypoxia experiments were carried out according to standard protocol of Hypoxyprobe-1 Plus Kit. Pimonidazole hydrochloride was subcutaneously injected at a dose of 60 mg/kg body weight at 1.5 h before the tumor excision. Hypoxia-positive signals in tumor tissues were detected by confocal fluorescence microscopy.

To detect the expression of HIF-1α, tumor slices were incubated with rabbit anti-HIF-1α antibody overnight at 4 °C, and then incubated with Alexa Fluor 647-labeled Goat Anti-Rabbit IgG (H + L) antibody for 2 h at room temperature. Cancer cell nuclei were stained with DAPI. HIF-1α-positive signals in tumors were obtained by confocal fluorescence microscopy.

### 2.11. In Vivo Combined Photothermal Brachytherapy

To investigate the combined therapeutic efficiency, mice with 125~150 mm^3^ of A549 tumors were randomly divided into six groups (n = 6 mice/group) for intratumoral injection of various agents: (i) Combined therapy group with the “multiple mild-hyperthermia” strategy. Tumors were injected with 15 μL of CuS-^131^I-PEG nanotheranostics (^131^I dose, 80 μCi/mouse), and then irradiated under the first NIR laser with a power of 1.5 W/cm^2^ at 15 min post-injection. Tumor temperature was raised to 39~40 °C and kept for 15 min, to alleviate tumor hypoxia. Two hours later, tumors were irradiated again under the second NIR laser with a power of 2.8 W/cm^2^, and tumor temperature was kept at 42~43 °C for 15 min in order to achieve photothermal brachytherapy. The above treatment session including mild hyperthermia stimulation and photothermal brachytherapy was carried out once every two days for three times. (ii) Combined therapy group without mild hyperthermia. Tumors were injected with 15 μL of CuS-^131^I-PEG nanotheranostics (^131^I dose, 80 μCi/mouse) and irradiated under 808 nm NIR laser with a power of 2.8 W/cm^2^ only at 2 h post-injection. Tumor temperature was raised to 42~43 °C and kept for 15 min to achieve the combined therapy. (iii) Free ^131^I-injected group. Tumors were only injected with 15 μL of free ^131^I (80 μCi/mouse). (iv) Blank control group with injection of PBS (15 μL) into tumor.

During NIR laser irradiation, mice were anesthetized with isoflurance inhalation (2%) and kept under anesthesia by isoflurane inhalation (1~1.5%). Meanwhile, mice were placed on a heating pat (35~36 °C) to keep the body warm and have them breathe normally. In addition, we measured in real time the temperature increase in the tumor area by infrared thermal camera in order to maintain the temperature in the specific range.

Each mouse was treated three times once every two days. Tumor size and body weight of mice were detected once every three days. However, from day17 to day23, authors were not admitted into the laboratory due to the spread of novel coronavirus in Shanghai, China, so that tumor size and body weight of mice were tested only once.

### 2.12. Statistical Analysis

Quantitative data were expressed as means ± S.D. (standard deviation). Student’s *t*-tests were used to evaluate the differences between groups. Statistical significance was indicated as * *p* < 0.05, ** *p* < 0.01 and *** *p* < 0.001.

## 3. Results

### 3.1. Preparation and Characterization of CuS-^131^I-PEG nanotheranostics

In this research, biodegradable and hydrophilic CuS-^131^I-PEG nanotheranostics were prepared based on the modified methods [10]. As shown in Figure 1A, ultra-small CuS nanoparticles were firstly prepared by our previous hydrothermal method [21]. Secondly, CuS was labeled by a radiopharmaceutical (^131^I) through the strong binding energy between Cu^2+^ and iodine ions to obtain ^131^I-labeled CuS (CuS-^131^I). Iodine ions were bound at the defect sites on the surface of CuS nanoparticles [11,22]. Finally, thiol PEG (PEG-SH) was modified on the surface of CuS-^131^I through the chelation reaction between thiol of PEG and metallic copper of CuS nanoparticles to obtain PEGylated CuS-^131^I (CuS-^131^I-PEG).

During preparation, the common iodine from NaI was a substitute for radionuclide ^131^I in order to facilitate the optimization of formulations. Firstly, PEG as the surface coating was crucial for the storage stability, radio-labeling stability, radiopharmaceutical metabolism, and so on. Coverage density of PEG had been systematically optimized by adjusting the ratio of PEG and CuS (as shown in Appendix A). In brief, when the mass of CuS was kept constant at 3.3635 mg, PEG amount was changed from 0.5 mg to 10 mg. Appendix A illustrated that the color of the PEGylated CuS-I solution gradually darkened from pale green to dark green along with the increase of PEG amount. Meanwhile, hydrodynamic size of CuS-I-PEG nanotheranostics also steadily increased as the PEG amount increased (in Appendix A). Moreover, the surface potential of CuS-I-PEG nanotheranostics had a measurable effect on physiochemical properties, such as stability. It was known that there was a greater repulsion force between nanoparticles when the surface potential of nanoparticles was more than 35 mV or less than −35 mV, leading to the strong stability of nanoparticles in solution. Seen from Appendix A, when the mass ratio of CuS to PEG was 3.3625:5, this CuS-I-PEG nanotheranostics with a surface potential of −37 mV was considered as the optimum formulation, which was used for the following experiments. Secondly, the molar ratio of NaI to CuS was also optimized. Appendix A showed that the amount of NaI had a small effect on the size and surface potential of nanoparticles.

Physicochemical properties of nanotheranostics were characterized in detail. As observed by transmission electron microscopy (TEM), CuS-I-PEG nanotheranostics had a spherical structure with a homogenous size distribution (in Figure 1B). The hydrodynamic diameter of CuS-I-PEG nanotheranostics was 21.56 ± 1.56 nm (in Figure 1C), which was favorable for their diffusion and uniform distribution in tumor tissues.

UV−Vis-NIR absorbance spectrum in Figure 1D illustrated that CuS-I-PEG nanotheranostics exhibited stronger optical absorption when the wavelength was in the range of 700 nm to 950 nm (Appendix A), which was similar to the absorption spectra of the published CuS nanoparticles [11]. Their stronger absorption to NIR light was advantageous for the better sensitivity and the higher signal-to-noise ratio of photoacoustic (PA) imaging.

Next, we substituted NaI with Na^131^I to obtain ^131^I-labeled nanotheranostics. The radio-labeling efficiency of ^131^I in CuS-^131^I-PEG nanotheranostics was about 70.4% after 2 h of reaction (Figure 1D). Moreover, a lower amount of ^131^I was detected from the obtained CuS-^131^I-PEG nanotheranostics after incubation in PBS and fresh mouse plasma at room temperature and 37 °C for 16 h (Figure 1E), respectively, indicating a better radio-labeling stability of ^131^I-CuS-PEG nanotheranostics in both PBS and mouse plasma.

### 3.2. Photothermal Properties of PEGylated Nanotheranostics

Photothermal nanoparticles as multifunctional nano-theranostics not only can achieve real-time diagnosis of tumor via photoacoustic imaging, but also can be used for photothermal therapy to inhibit tumor growth and metastasis [23,24,25]. In this research, CuS-I-PEG nanotheranostics exhibited the concentration-dependent photothermal heating effect (Figure 2A,B) and had a stronger photothermal stability upon 808 nm NIR laser irradiation on/off for three cycles (Figure 2C). Photothermal conversion efficiency (PCE) of CuS-I-PEG was 78.4% (Figure 2D), which was higher than that of gold nanoparticles [22].

Due to the higher PCE (in Figure 2D) of CuS-I-PEG nanotheranostics in NIR region, we investigated their photothermal heating effect at tumor tissues in vivo. A549 tumors on mice were intratumorally injected with 10 uL of CuS-I-PEG solution containing 2.5 mmoL of CuS. Then, tumors were irradiated by 808 nm NIR laser with a power density of 2.8 W/cm^2^. The temperature evolution was detected by infrared thermal camera. As expected, tumors injected with CuS-I-PEG nanotheranostics were quickly heated with the temperature maintained at 45 °C under the 808 nm NIR laser irradiation. In contrast, tumors injected with PBS had a lower temperature increase.

### 3.3. Cell Cytotoxicity of CuS-^131^I-PEG Nanotheranostics In Vitro

Potential effect of CuS-^131^I-PEG nanotheranostics on cellular viability under hypoxic conditions or normal cell culture conditions was investigated through Cell Counting Kit-8 (CCK-8) assay 24 h and 48 h post-treatment, respectively. CoCl_2_ was used to induce the hypoxic status of A549 cells (as shown in Figure 3A). Results in Figure 3B showed that cell viability under hypoxic conditions was much higher than that under normal cell culture conditions after incubation with CuS-^131^I-PEG at the same concentration of nanotheranostics and at the same radiation dose. These results indicated that cell apoptosis was suppressed under hypoxic conditions and that DNA double strands also could not be effectively broken by ^131^I-mediated radiotherapy. Moreover, as shown in Appendix A, cell viability of A549 cells after incubation with free CuS-I-PEG nanotheranostics was still higher than 80% even at a much higher concentration of 256 μg/mL, suggesting the better biocompatibility of CuS-I-PEG nanotheranostics.

### 3.4. Intratumoral Distribution of CuS-^131^I-PEG Nanotheranostics by Photoacoustic Imaging

Uniform distribution of CuS-^131^I-PEG nanotheranostics in tumor tissues is highly associated with the efficiency of the combined photothermal brachytherapy. In this research, we proposed that mild hyperthermia could promote the intratumoral distribution of nanotheranostics. To confirm this hypothesis, A549 tumor-bearing mice were randomly separated into two groups: (i) Non-irradiated group, i.e. “intratumoral injection of CuS-^131^I-PEG nanotheranostics without 808 nm NIR irradiation”; (ii) NIR-irradiated group, i.e. “intratumoral injection of CuS-^131^I-PEG nanotheranostics with 808 nm NIR irradiation”, as shown in Figure 4A.

In the non-irradiated group, CuS-^131^I-PEG nanotheranostics were distributed only at the periphery of tumor tissues instead of the whole tumors even at 2 h post-injection (Appendix A). By contrast, in the NIR-irradiated group, tumors were irradiated for 20 min at 1.5 W/cm^2^ after intratumoral injection of CuS-^131^I-PEG nanotheranostics so that the temperature at the tumor was raised to 39~40 °C and kept for 20 min. At 0.5 h post-irradiation, PA red-positive signals (in Figure 4B,C) were nearly distributed around the whole tumor tissues, suggesting that NIR-triggered mild hyperthermia was conductive to the better diffusion of CuS-^131^I-PEG nanotheranostics inside tumors. However, the intratumoral uniformity of PA-positive signals at 1 h and 2 h post-irradiation was not better than that at 0.5 h post-irradiation, suggesting that CuS-^131^I-PEG nanotheranostics possibly leaked into the surrounding tissues by lymphatic drainage, which was consistent with those of others [10]. In all, our results indicated that (i) mild-hyperthermia promoted the quick diffusion and uniform distribution of CuS-^131^I-PEG nanotheranostics in the solid tumor; (ii) the best time with uniform intratumoral distribution of CuS-^131^I-PEG nanotheranostics was 0.5 h post-intratumoral injection.

### 3.5. Intratumoral Accumulation of ^131^I in CuS-^131^I-PEG Nanotheranostics Monitored by SPECT/CT Imaging

Radionuclide ^131^I emitting strong beta and gamma rays not only could serve as a radiopharmaceutical for anti-tumor brachytherapy but could also be considered as a diagnostic reagent for nuclear imaging, i.e. SPECT/CT imaging [26,27]. In this study, to evaluate the tumor accumulation of nanotheranostics with NIR-triggered mild hyperthermia, A549-bearing mice were scanned by SPECT/CT imaging after intratumoral injection of free ^131^I and CuS-^131^I-PEG nanotheranostics (Figure 5A), respectively. Results in Figure 5B showed that radioactive signals at tumor locations after injection of free ^131^I entirely disappeared at 1.5 h post-injection, indicating poorer tumor accumulation capability and faster metabolism of free ^131^I.

In CuS-^131^I-PEG injected mice, tumors on mice were irradiated by 808 nm NIR laser after intratumoral injection so that the tumor temperature was raised to 39~40 °C and kept constant for 15 min. Radioactive signals in tumors were clearly detected even at 6.0 h post-injection of CuS-^131^I-PEG (Figure 5B,C), suggesting an excellent accumulation of CuS-^131^I-PEG at tumor sites compared to free ^131^I. This accumulated duration (more than 6.0 h) of CuS-^131^I-PEG nanotheranostics in tumor tissues was enough to kill cancer cells by irradiated beta rays of radionuclide ^131^I. Meanwhile, the vast majority of radio-pharmaceutical (^131^I) in CuS-^131^I-PEG nanotheranostics was excreted from the body within 6 h via urine and feces from kidney and liver (Figure 5B), respectively, to avoid the radioactive damage to normal tissues and cells.

### 3.6. In Vivo Anti-Tumor Efficiency of the Combined Photothermal Brachytherapy

To evaluate the effect of “multiple mild-hyperthermia” strategy on the anti-tumor outcome, we conducted multiple mild-hyperthermia to realize the long-term alleviation of tumor hypoxia. The experimental flow chart is illustrated in Figure 6A. In the combined therapy group without mild-hyperthermia, although tumors showed partially delayed growth at the initial stage, they grew up very quickly at the later time points, indicating that the hypoxia-associated tumor cells would not be effectively killed. In the free ^131^I-injected group, tumors on mice also showed striking growth. As expected, in the combined therapy group with multiple mild-hyperthermia stimulation, the growth of tumors was completely inhibited. This was likely attributed to the fact that long-term alleviation of tumor hypoxia was successfully achieved by multiple mild-hyperthermia stimulation in tumor tissues, leading to the long-term retention of radiation sensitivity. Collectively, the “multiple mild-hyperthermia” strategy indeed enhanced the efficiency of combined photothermal brachytherapy in in-vitro (Figure 3B) and in-vivo (Figure 6B) studies.

In addition, the body weights of the mice was simultaneously monitored to further evaluate whether these treatments had any toxic effect. Results in Figure 6C showed that the body weights of mice in all treated groups were equivalent to those in the control group (PBS group), indicating good safety and biocompatibility of our developed CuS-^131^I-PEG nanotheranostics.

### 3.7. Mechanism of Tumor Hypoxic Alleviation through Mild Hyperthermia

After demonstrating the efficiency of photothermal brachytherapy of CuS-^131^I-PEG nanotheranostics, we would like to illustrate the anti-tumor mechanism of the combined therapy under the “multiple mild-hyperthermia” strategy. A549 tumors on mice were intratumorally injected with CuS-I-PEG nanotheranostics and irradiated by 808 nm NIR laser to mildly increase the tumor temperature to 39~40 °C at 15 min, 22 h post-injection, respectively. In control groups, tumors were not irradiated by NIR laser or were irradiated by single NIR laser only at 15 min post-injection. Tumors in all groups were collected at 2 and 24 h post-injection. The results of immunofluorescence staining using pimonidazole-based hypoxyprobe showed that hypoxic signals (green color in Figure 7, left) were weakened after each round of NIR laser irradiation in the double NIR laser irradiation group. However, in the single NIR laser irradiation group, the hypoxic state was gradually recovered and reached its initial state at 24 h after NIR laser irradiation. That is, double mild hyperthermia at local tumor tissues indeed repeatedly promoted oxygen levels inside tumor tissues at 24 h, leading to re-alleviate tumor hypoxia.

Besides pimonidazole-based hypoxia staining, the hypoxia-inducible factor-1α (HIF-1α) expression level in tumors was also evaluated by immunofluorescence staining. HIF-1α was a transcription factor that was highly expressed in hypoxic cells [28,29]. The HIF-1α expression level was closely associated with occurrence, growth, angiogenesis, local infiltration and distal metastasis of lung cancers [30,31]. Many research studies reported that over-expression of HIF-1α enhanced the DNA damage repair and angiogenesis, finally leading to the hypoxia-induced radio-resistance [32,33]. In this study, we found that HIF-1α expression level (red color in Figure 7, right) was significantly suppressed at 2 h after NIR laser irradiation, but it recovered to the initial level at 24 h post-irradiation. In the double mild hyperthermia group, HIF-1α expression could be prolonged at a lower level even at 24 h. These results indicated that long-term lower expression of HIF-1α was beneficial for suppressing the DNA damage repair and angiogenesis, further reversing the hypoxia-induced radio-resistance, which was consistent with results in Figure 6B. In all, both increasing oxygen levels and suppressing HIF-1α expression inside tumor tissues was simultaneously accomplished after multiple mild-hyperthermia stimulation based on the NIR-irradiated CuS-^131^I-PEG nanotheranostics.

## 4. Discussion

To alleviate tumor hypoxia and reverse hypoxia-associated radio-resistance, many methods have been developed in clinical settings, such as normobaric oxygen breathing, oxygen delivery via hyperbaric oxygen and blood transfusion, etc. However, these routine clinical practices of hypoxic modification are rather restricted and inconclusive because of their difficult techniques and underpowered outcomes in routine practices [34]. In recent years, researchers have explored novel strategies to reverse tumor hypoxia in brachytherapy through the advancement of nanotechnology. For example, one strategy is to use nanoparticles (e.g. semiconducting polymer-stabilized perfluorecarbon [35] and perfluorocarbon-loaded hollow Bi_2_Se_3_ nanoparticles [36], etc.) as oxygen shuttles to directly deliver oxygen to tumor tissues. These nanoparticles possess better properties of oxygen-loading capacity, controllable oxygen release rate and higher alleviation efficiency of tumor hypoxia [37]. The loaded oxygen could be transported to the tumor sites alongside nanoparticles. However, oxygen-shuttered nano-platforms need to involve the stringent and complicated experimental conditions. The other strategy is to apply the oxygen-generating nanomaterials (e.g., gold nanoclusters [38], manganese dioxide nanoparticles [39], etc.) to decompose hydrogen peroxide into oxygen. However, their oxygen-generating efficiency is restricted by hydrogen peroxide concentration in tumor tissues. Compared to the above methods, our developed “multiple mild-hyperthermia” strategy had great advantages, including simplicity, repeated relief of tumor hypoxia, and spatiotemporal controllability.

Recently, mild hyperthermia has been explored for alleviating tumor hypoxia. For example, Meng et. al. have reported that NIR-triggered in situ gelation system can relieve the tumor hypoxia under NIR irradiation and kill cancer cells by synergistic photothermal brachytherapy [11]. Atkinson et. al. have found that local hyperthermia induced by gold nanoshells can sensitize cancer stem cells to radiation therapy [40]. However, tumor tissues normally recover their hypoxic state at about 24 h after the stimulation of mild hyperthermia, again leading to the hypoxia-associated radio-resistance [11]. Therefore, multiple mild-hyperthermia was greatly necessary to long-term overcome tumor hypoxia and long-term maintain radio-sensitivity.

Moreover, multiple mild-hyperthermia also demonstrated two functions in our study. The first was to promote the intratumoral distribution of nanotheranostics. Uniform distribution of nanotheranostics in tumors also was helpful in achieving better photothermal therapy. The second was to enhance blood flow, improve oxygen levels and alleviate tumor hypoxia, which was favorable for higher brachytherapy. In addition, multiple mild-hyperthermia may change the permeability of biological membranes or increase the amount of reactive oxygen species (ROS). That is, these factors including oxygen levels, biological membrane permeability and ROS amount, would be simultaneously influenced by NIR laser irradiation. In the future, we need to explore effects of biological membrane permeability and ROS on the tumor hypoxic status and anti-tumor efficiency.

## 5. Conclusions

In conclusion, we designed a biodegradable and biocompatible nanotheranostics, CuS-^131^I-PEG with a “multiple mild-hyperthermia” strategy, for alleviating tumor hypoxia and reversing hypoxia-associated radio-resistance under NIR laser irradiation. In this nanotheranostics, ^131^I emitting stronger beta and gamma rays were considered as theranostics agents for SPECT/CT imaging and internal brachytherapy. Ultra-small CuS nanoparticles with higher photothermal conversion stability also served as diagnostic and therapeutic reagents for photoacoustic imaging and photothermal therapy. ^131^I was easily embedded into defect sites of CuS to obtain ^131^I-labeled CuS nanoparticles (CuS-^131^I). A PEG polymeric matrix as a surface coating was modified on the surface of CuS-^131^I, to improve the stability and tumor retention of ^131^I. After the intratumoral injection of CuS-^131^I-PEG nanotheranostics, tumors were irradiated by the 1st NIR laser to mildly increase tumor temperatures to 39~40 °C. According to immunofluorescence staining, this mild hyperthermia in tumors could elevate the intratumoral oxygen level and alleviate the tumor hypoxia, leading to reverse radio-resistance. After 2 h of the 1st irradiation, tumors were irradiated again by NIR laser to achieve the combined photothermal brachytherapy. In the whole treatment course, tumors were treated three times by the combined photothermal brachytherapy. Mild hyperthermia (39~40 °C) was carried out 2 h in advance of each treatment. That is, there were three times for mild hyperthermia stimulations, which was considered as a “multiple mild-hyperthermia” strategy. Notably, the growth of tumors was completely inhibited in the combined therapy group with the “multiple mild-hyperthermia” strategy compared with that in the group without mild hyperthermia. Considering the simple and effective strategy of multiple mild-hyperthermia, our developed CuS-^131^I-PEG nanotheranostics could have significant potential in the application of cancer combined therapies.

## Figures and Tables

**Figure 1 pharmaceutics-14-02669-f001:**
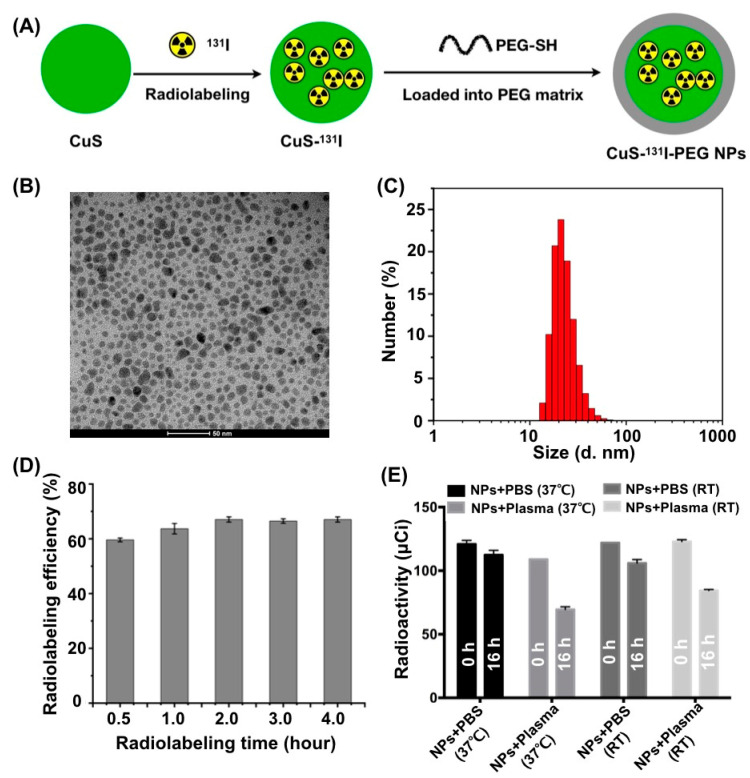
Preparation and physicochemical properties of CuS-^131^I-PEG nanotheranostics. (**A**) Schematic illustration to show the preparation of CuS-^131^I-PEG nanotheranostics. (**B**) TEM image of nanotheranostics. (**C**) Hydrodynamic size distribution of nanotheranostics measured by dynamic light scattering. (**D**) Radiolabeling efficiency of ^131^I at different time points. (**D**) Radiolabeling efficiency of ^131^I in CuS-^131^I-PEG nanotheranostics for 4 h. (**E**) Radiolabeling stability of CuS-^131^I-PEG nanotheranostics incubated with fresh mouse plasma and PBS at 37 °C and room temperature (RT) for 16 h, respectively.

**Figure 2 pharmaceutics-14-02669-f002:**
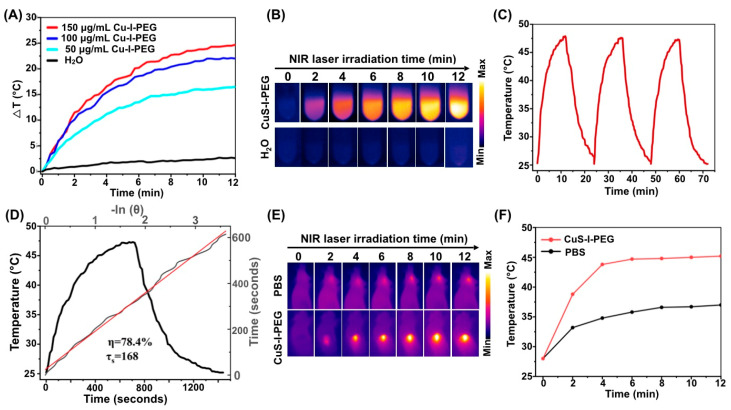
Photothermal conversion properties of CuS-^131^I/I-PEG nanotheranostics. (**A**) The temperature changes of CuS-I-PEG nanotheranostics under 808 nm NIR laser irradiation at a power of 2.8 W/cm^2^. (**B**) Photothermal conversion images of (**A**). (**C**) Photothermal stability of CuS-I-PEG nanotheranostics upon 808 nm laser on/off for three cycles at a power of 2.8 W/cm^2^. (**D**) Photothermal heating/cooling curve (in black) of CuS-I-PEG nanotheranostics in PBS under 808 nm NIR laser irradiation at a power of 2.8 W/cm^2^, and (gray line) liner time data versus −ln (θ) obtained from the cooling period of black line. Red line represented the fitting result of gray line. (**E**) Photothermal conversion images of CuS-I-PEG nanotheranostics in tumor tissues on mice under 808 nm NIR laser irradiation at a power of 2.8 W/cm^2^. (**F**) Temperature changes of CuS-I-PEG nanotheranostics corresponding to (**E**).

**Figure 3 pharmaceutics-14-02669-f003:**
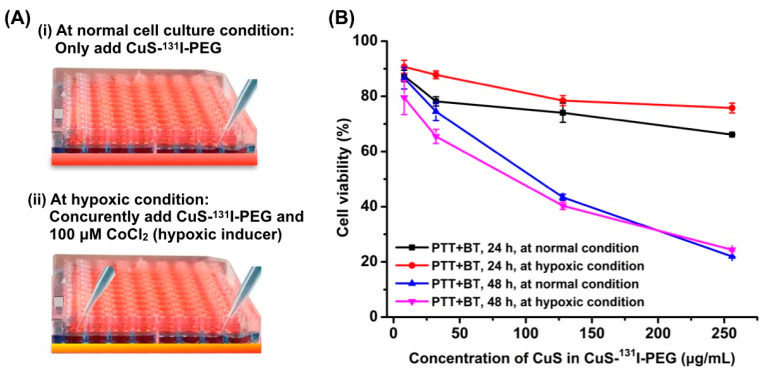
In vitro cytotoxicity of photothermal brachytherapy in A549 cells by CCK8 assay. (**A**) Schematic illustration of two different cell culture conditions. (**B**) In vitro cell viability of CuS-^131^I-PEG nanotheranostics under NIR laser irradiation under normal cell culture conditions and hypoxic conditions, respectively.

**Figure 4 pharmaceutics-14-02669-f004:**
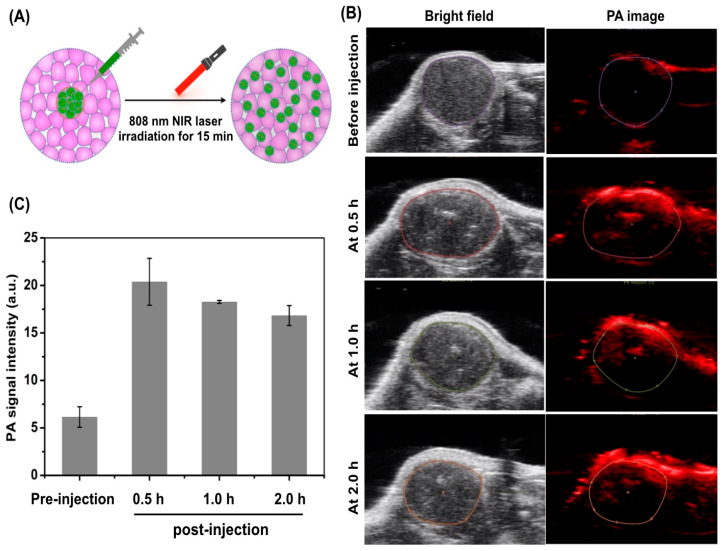
Intratumoral distribution of CuS-^131^I-PEG nanotheranostics in vivo. (**A**) Schematic illustration of intratumoral distribution of CuS-^131^I-PEG nanotheranostics after mild hyperthermia stimulation through 808 nm NIR laser irradiation for 15 min. (**B**) Photoacoustic signals of distribution of CuS-^131^I-PEG nanotheranostics inside tumors for 2.0 h. (**C**) Quantitative analysis of photoacoustic signal intensity of ^131^I in (**B**).

**Figure 5 pharmaceutics-14-02669-f005:**
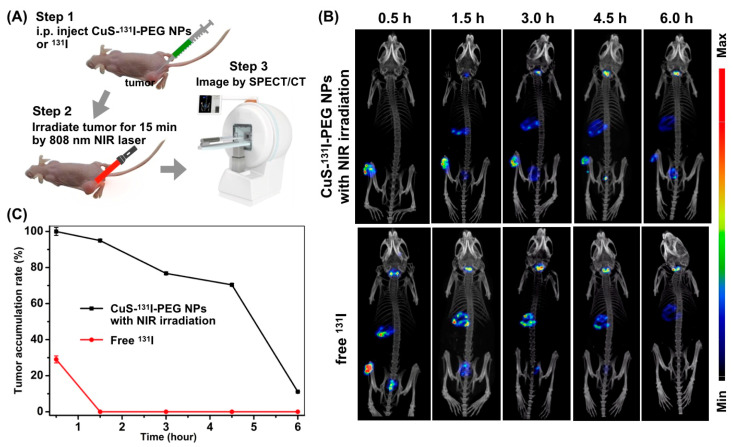
SPECT images for intratumoral accumulation and metabolic process of 131I after intratumoral injection of CuS-131I-PEG nanotheranostics and 131I, respectively. (**A**) Schematic illustration of process in tumor accumulation study. (**B**) SPECT images of 131I after intratumoral injection of CuS-131I-PEG nanotheranostics and ^131^I for 6.0 h, respectively. (**C**) Quantification of tumor accumulation rate of ^131^I.

**Figure 6 pharmaceutics-14-02669-f006:**
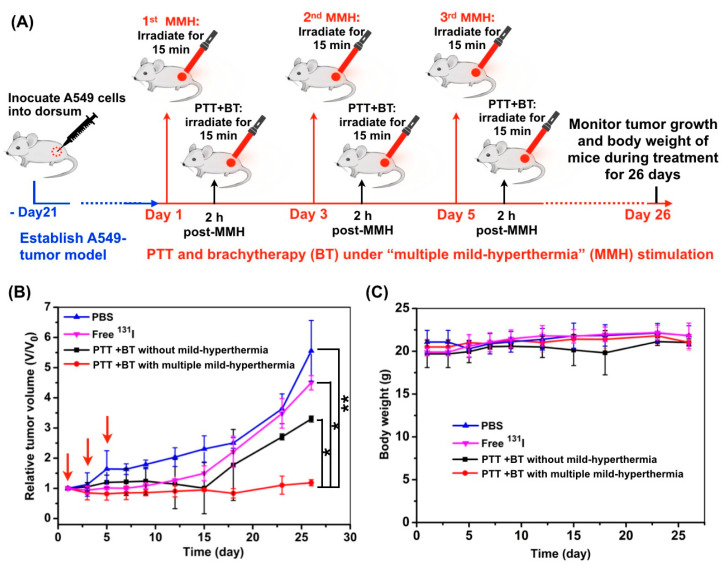
In vivo anti-tumor efficiency of photothermal brachytherapy. (**A**) Scheme of the combined photothermal brachytherapy under multiple mild-hyperthermia stimulation in A549 tumor-bearing mice. (**B**) Tumor growth curves after treatment with PBS, free ^131^I, photothermal brachytherapy (PTT + BT) without mild hyperthermia, and photothermal brachytherapy (PTT + BT) with multiple mild-hyperthermia, respectively. Red arrows indicated the time points for mild-hyperthermia stimulation. (**C**) Changes of body weight of mice. * *p* < 0.05, ** *p* < 0.01.

**Figure 7 pharmaceutics-14-02669-f007:**
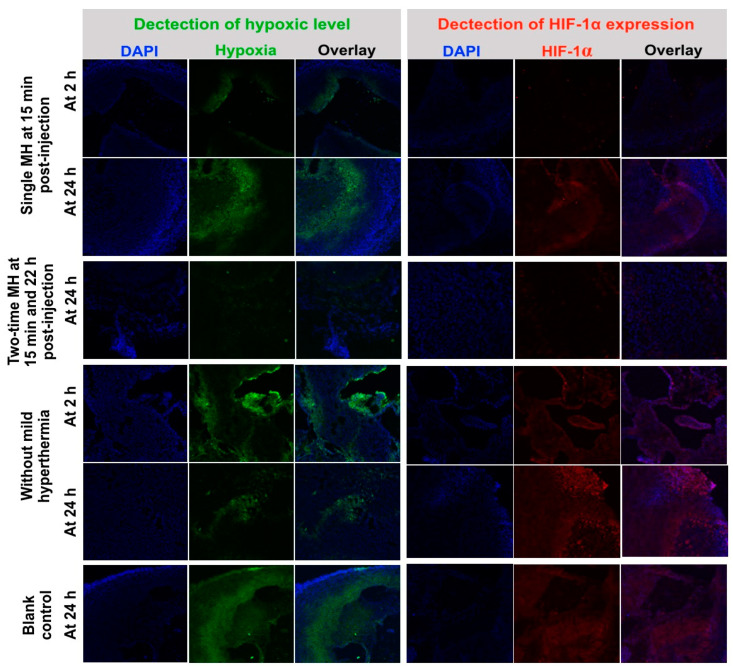
Alleviation mechanism of tumor hypoxia under mild hyperthermia (MH) stimulation. Representative immunofluorescence images of pimonidazole-stained hypoxic signals (green) in left column, and HIF-1α antibody-stained hypoxic signals (red) in right column. Nuclei were stained with DAPI (blue).

## Data Availability

Not applicable.

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
