# Peer review of "CuS-131I-PEG Nanotheranostics-Induced “Multiple Mild-Hyperthermia” Strategy to Overcome Radio-Resistance in Lung Cancer Brachytherapy"

_pharmaceutics, 2022, doi:10.3390/pharmaceutics14122669_

Round 1
Reviewer 1 Report
The manuscript describes a well-conducted investigation on the effects of the photothermal effect of NIR laser irradiation as a complementary technique to be used in antitumor treatments. This topic is becoming more relevant as new complementary techniques are needed to treat tumors that do not respond to conventional treatments.
I recommend its publication, but after clarifying some minor point:
1) The authors irradiated mice several times using a NIR Laser device, but it is not specified in the text how they carry out this irradiation (section 2.7 in Materials and Methods lacks this information). Did the authors anesthetize the mice, and if so, what method did they use? (The effect of NIR laser irradiation is mainly thermal as they claim, and anesthetic treatment could decrease body temperature and oxygen supply to tumor tissues).
Did the authors record any harmful effects on the skin irradiated with the NIR Laser?
Did the authors record any deleterious effects of mild hyperthermia in non-tumor mouse tissues?
2) Did the authors employ any cooling system (to surrounding non tumoral tissues) during NIR laser irradiation in mice?
3)The legend of Figure 1.E says that the NPS were incubated with water to assess their stability. But in the text (lines 313, 314 and 315) it is mentioned that PBS was used, what was the medium used?
4) Figure 3, shows in vitro effects of NPs in a tumor line (A549). did the authors register in vitro toxicity upon non-tumor lines?
5) In line 449 a typographical error can be pointed out to be corrected. It should be "In vitro" instead of "in intro".
6) The authors point to the increase in oxygen delivery as the only effect induced by mild hyperthermia after NIR irradiation. Other effects are not discussed, such as changes in the permeability of biological membranes or increase in reactive oxygen species due to the NIR irradiation itself (upon internal chromophores). This discussion should be mentioned in the conclusions section.
Author Response
Dear Reviewer,
We sincerely thank the reviewer for your constructive suggestions and helpful comments that help to improve the presentation of our manuscript. We have carefully considered your comments and revised our manuscript. We believe that our responses can address all your concerns. Please see the attachment.
Yours sincerely,
Zeyu Xiao

Reviewer 2 Report
1. References:
5 - no bibliographic data
6 - no title and authors (!?)
2. Zhang1 - no superscript
3. "Lung cancer as a kind of the most malignancies has higher morbidity and mortality" - in relation to what?
4. "These nanoparticles possess better properties of a higher oxygen-loading capacity, controllable oxygen release rate and higher alleviation efficiency of tumor hypoxia." - How do nanoparticles transport oxygen? No literature justification.
5. Work has a disturbed structure. In the opinion of the reviewer, the second part of the introduction should be transferred to the discussion. Balance of references is also unacceptable. More than 25 citations (out of 32) are used in the introduction.
6. The reviewer finds the work interesting, but requires a thorough development of part of the discussion. The results should be better related to the leading literature. There are many important and omitted studies that correspond to the results obtained.
Author Response
Dear Reviewer,
We sincerely thank the reviewer for your helpful comments that help to improve the presentation of our manuscript. According to your constructive suggestions, we have carefully revised our manuscript. We believe that our responses can address all the concerns of the reviewer well. Please see the attachment.
Sincerely yours,
Zeyu Xiao
